# A Large Family with p.Arg554His Mutation in *ABCD1*: Clinical Features and Genotype/Phenotype Correlation in Female Carriers

**DOI:** 10.3390/genes12050775

**Published:** 2021-05-19

**Authors:** Rosa Campopiano, Cinzia Femiano, Maria Antonietta Chiaravalloti, Rosangela Ferese, Diego Centonze, Fabio Buttari, Stefania Zampatti, Mirco Fanelli, Stefano Amatori, Carmelo D’Alessio, Emiliano Giardina, Francesco Fornai, Francesca Biagioni, Marianna Storto, Stefano Gambardella

**Affiliations:** 1IRCCS Neuromed, 86077 Pozzilli, Italy; rosacampopiano85@gmail.com (R.C.); Cinzia.femiano@neuromed.it (C.F.); maria.ant87@libero.it (M.A.C.); ferese.rosangela@gmail.com (R.F.); centonze@uniroma2.it (D.C.); fabio.buttari@neuromed.it (F.B.); stefania.zampatti@gmail.com (S.Z.); Carmelo.dalessio@neuromed.it (C.D.); francesco.fornai@med.unipi.it (F.F.); frbiagioni@libero.it (F.B.); marianna.storto@neuromed.it (M.S.); 2Laboratory of Synaptic Immunopathology, Department of Systems Medicine, Tor Vergata University, Via Montpellier 1, 00133 Rome, Italy; 3Genomic Medicine Laboratory, IRCCS Fondazione Santa Lucia, 00179 Rome, Italy; emiliano.giardina@uniroma2.it; 4Department of Biomolecular Sciences, University of Urbino “Carlo Bo”, 61029 Urbino, Italy; mirco.fanelli@uniurb.it (M.F.); Stefano.amadori@uniurb.it (S.A.); 5Department of Biomedicine and Prevention, University of Rome “Tor Vergata”, 00133 Rome, Italy; 6Department of Translational Research and New Technologies in Medicine and Surgery, University of Pisa, 56124 Pisa, Italy

**Keywords:** neurogenetics, X-linked adrenoleukodystrophy, *ABCD1*, next generation sequencing, diagnosis

## Abstract

X-linked adrenoleukodystrophy (X-ALD, OMIM #300100) is the most common peroxisomal disorder clinically characterized by two main phenotypes: adrenomyeloneuropathy (AMN) and the cerebral demyelinating form of X-ALD (cerebral ALD). The disease is caused by defects in the gene for the adenosine triphosphate (ATP)-binding cassette protein, subfamily D (*ABCD1*) that encodes the peroxisomal transporter of very-long-chain fatty acids (VLCFAs). The defective function of *ABCD1* protein prevents β-oxidation of VLCFAs, which thus accumulate in tissues and plasma, to represent the hallmark of the disease. As in many X-linked diseases, it has been routinely expected that female carriers are asymptomatic. Nonetheless, recent findings indicate that most *ABCD1* female carriers become symptomatic, with a motor disability that typically appears between the fourth and fifth decade. In this paper, we report a large family in which affected males died during the first decade, while affected females develop, during the fourth decade, progressive lower limb weakness with spastic or ataxic-spastic gait, tetra-hyperreflexia with sensory alterations. Clinical and genetic evaluations were performed in nine subjects, eight females (five affected and three healthy) and one healthy male. All affected females were carriers of the c.1661G>A (p.Arg554His, rs201568579) mutation. This study strengthens the relevance of clinical symptoms in female carriers of *ABCD1* mutations, which leads to a better understanding of the role of the genetic background and the genotype-phenotype correlation. This indicates the relevance to include *ABCD1* genes in genetic panels for gait disturbance in women.

## 1. Introduction

X-linked adrenoleukodystrophy (X-ALD, OMIM #300100) is the most common peroxisomal disorder. The disease is caused by defects in the gene for the adenosine triphosphate (ATP)-binding cassette protein, subfamily D (*ABCD1*) [1,2,3,4]. This gene encodes the peroxisomal transporter of very-long-chain fatty acids (VLCFAs) [5,6]. The defective function of *ABCD1* protein prevents β-oxidation of VLCFAs, leading to accumulation in tissues and plasma, which represents the disease hallmark [7].

X-ALD is clinically characterized by two main phenotypes: adrenomyeloneuropathy (AMN) and the cerebral demyelinating form of X-ALD (cerebral ALD) [1,8,9].

Cerebral ALD presents usually with a rapidly progressive inflammatory demyelination of the cerebral white matter, which produces severe cognitive and motor disability in childhood (childhood cerebral ALD; CCALD) and adolescence (adolescent cerebral ALD) [9].

The pathology of AMN is distinct from cerebral ALD and it is mostly characterized by a non-inflammatory distal axonopathy, which mostly affects the long descending pathways in the white matter of the spinal cord, with an emphasis on the corticospinal pathway, which leads to progressive spastic para-paresis [1,8,9].

As in many X-linked diseases, it was assumed that female carriers remain asymptomatic until it was reported that also heterozygous females eventually, as time goes by, suffer from AMN symptoms [10,11,12,13]. Moreover, these female patients, even in the early asymptomatic stage, already carry some abnormalities, which can be detected by instrumental neurophysiology [5]. Sometimes, at this stage also adrenal failure and cerebral adrenoleukodystrophy may appear [5,14,15,16,17].

Recent clinical evaluations in prospective cross-sectional cohort studies have detected neurological impairment in females, reporting the evolution of phenotypes with aging [5,18]. These studies demonstrated that carriers develop an adrenomyeloneuropathy-like phenotype with a strong association between symptomatic status and age [5]. Symptoms include myelopathy (63%) and/or peripheral neuropathy (57%) [5].

Generally, the clinical phenotype of carrier women is milder than in men, with a typical onset after the fourth decade. However, some exceptions have been reported and sometimes a severe phenotype can be recognized in carrier women with early-onset [19].

Based on mutational analysis of the *ABCD1* gene in X-ALD patients, about 600 different mutations have been described so far [20] (http://www.x-ald.nl, accessed on 10 October 2020).

According to the known intra-familial variability, it was supposed that other (epi)genetic factors contribute to the phenotype of patients with *ABCD1* mutations [21].

Therefore, although large deletions, nonsense or frameshift mutations that result in the complete absence of a functional ALDP have been found in patients covering the full spectrum of X-ALD phenotypes, no strong genotype-phenotype correlation has been described. In clinical practice, the absence of a strong genetic relationship with the phenotype makes difficult the neurological characterization, leading often to misdiagnosis.

In this paper, we report on a large family in which affected males died during the first decade, while affected females developed symptoms within the fourth decade with spastic or ataxic-spastic gait, tetra-hyperreflexia, lower limbs hyposthenia, and sensory complaints.

## 2. Materials and Methods

### 2.1. Ethics

A written informed consent for genetic analyses was obtained from the patients or patients’ parents. The research work was carried out following ethical principles and the Italian legislation. The study was approved by IRCCS Neuromed ethical committees. The study is registered on ClinicalTrials.gov (NCT03084224).

### 2.2. Demographic and Clinical Features

This paper describes a five-generation family in which two main phenotypes of X-ALD are segregated (Figure 1). The affected male died in the first decade (III:4; III:8; III:13; and III:14), and all affected females present with gait disturbance and different neurological features (Figure 1, Table 1). The genetic evaluation included nine people, including eight females and one male. Of the eight females, five are affected (III:2; III:6; III:10; IV:2; and IV:4), age at onset ranging between 34 and 42 years (average 38 years); and 3 are healthy (IV:8; V:1; and V:2), age at evaluation ranging between 26 and 39 years (average 31 years). The healthy male IV:5 is 35 years old.

Clinical, neurological examination and brain MRI scan were performed in the affected female (Table 1, Figure 2). A complete neuropsychological evaluation was performed in patients IV:2 and IV:4 and a psychological interview in patient III:3.

### 2.3. Genetic Analysis

Genomic DNA was isolated from peripheral blood leukocytes according to standard procedures (QIAamp DNA Blood Mini Kit–QIAGEN). Clinical exome sequencing considering about 5000 human genes was performed using the Clinical Exome Solution kit (Sophia Genetics, SA, Boston, MA, USA), following the manufacturer’s instructions. The resulting libraries were processed for paired-end sequencing on the MiSeq platform Illumina (San Diego, CA, USA). Sophia DDM^®^ platform (Sophia Genetics, SA) was used for automated annotation, characterization, and selection of potentially pathogenic variants. Direct evaluation of the data sequence was performed by the Integrative Genomics Viewer v.2.3.

A second analysis by using GenomeUp platfarm was performed (https://platform.genomeup.com/, accessed on 10 October 2020) using the Best Practices workflows of GATK v4.1 for germline variant calling.

Potentially pathogenic variants were interpreted according to ACMG criteria [22]. ACMG classification was compared with automatic classification performed by Varsome genome interpreter (https://varsome.com/, accessed on 10 October 2020).

Variant NM_000033:c.[1661G>A]; NP_000024.2:p.(Arg554His) rs201568579 (ClinVar # VCV000166625) in *ABCD1* gene was selected as potentially responsible for the clinical phenotype (class 5, PP5-PM1-PM2-PM5-PP2-PP3).

Mutation re-sequencing and segregation analysis were performed by Sanger sequencing ABI 3130xl Genetic Analyzer (Applied Biosystems, Foster City, CA, USA). To perform direct re-sequencing of the pathogenic variant in *ABCD1*, sequence-specific primers were designed (FW–5′ GAGTATCTTGGGGGAGGCAG 3′; RW–5′ ATCTGTGTGGTGTTGGTCCTC 3′) to avoid amplification of homolog sequences of the *ABCD1* pseudogenes at chromosomes 2p11, 10p11, 16p11, and 22q11 (92–96% of similarities with exons 7–10 of *ABCD1* gene).

### 2.4. Biochemical Analysis

Blood samples were collected to measure plasma levels of the VLCFA C22:0, C24:0, and C26:0. VLCFA values of C26:0, of C26:0/C22:0, and C24:0/C22:0 were evaluated. Details of all these methods were described in Habekost et al., 2014 [18].

### 2.5. Literature Review of p.(Arg554His) Variant

A systematic review of the literature was conducted to identify studies reporting male and female patients with p.(Arg554His) mutation. The literature search identified 14 publications describing 23 patients. The gender is reported for 13 male patients. The phenotype reported in association with this variant includes ALD, CCALD, AMN, and asymptomatic carrier (Table 2).

### 2.6. Statistic and Data Analysis

An unpaired *t*-test was conducted comparing reference values with plasma values of C26:0, C26:0/C22:0, and C26:0/C24:0 ratio in female carriers vs. healthy male.

## 3. Results

Probands III:2 were referred to the Neurology Unit of Neuromed IRCCS (Italy), for gait difficulties, imbalance, and lower limb stiffness. Familial anamnesis revealed that she is part of a large family in which affected males died with the first decade, while affected females showed progressive spastic or ataxic-spastic gait, tetra-hyperreflexia, lower limbs hyposthenia, and sensory complaints.

Sequencing analysis performed on III:2 identified a pathogenic variant in *ABCD1* gene (NM_000033:c.[1661G>A]; NP_000024.2:p.(Arg554His); rs201568579). This variant is responsible for X-linked adrenoleukodystrophy (OMIM #300100), and it is has been described in male patients affected by X-ALD.

To fully understand the role of this variant in the clinical phenotype of the numerous affected patients, genetic analyses were performed in four more symptomatic females (III:6; III:10; IV:2; IV:4), confirming the segregation of the identified mutation with the pathological phenotype. To further confirm the segregation, a sequencing analysis on a healthy son (IV:5) of an affected woman (III:6) was performed. The analysis revealed the absence of mutation, confirming the inheritance of the maternal wild-type allele.

VLFA dosage was conducted in patients III:2; III:6; III:10; IV:2; IV:4; and IV:8. Plasma C26:0 levels, C24:0/C22:0, and C26:0/C22:0 ratios were increased in all the female samples (Table 1, C26:0 levels *p* = 0.0078; C26:0/C22:0 *p* = 0.0127).

The female symptomatic carriers (III:2; III:6; III:10; IV:2 and IV:4) were clinically characterized (Table 1).

The age of symptoms ranged between 34 and 42 years, with an average of 38 years. All the women showed first progressive lower limb weakness with gait difficulties, imbalance, sometimes lower limb stiffness and, in one case, cramps or paraesthesia. Sphincter disturbances were present in varying degrees. Neurological examination showed spastic or ataxic-spastic gait, tetra-hyperreflexia, Babinski sign, lower limbs hyposthenia, sensory complaints (lower limb hypopallesthesia or pain), and in some cases diplopia or light upper limb dysmetria. No patients had hyperpigmentation or another adrenal symptom of adrenal failure. All of them showed multiple discopathy of the spinal cord without myelopathy.

Complete cognitive profile was assessed in patients IV:2 and IV:4. The first showed a slight impairment in praxic-constructive and logical-deductive skills, severe depression, and anxiety with conversion aspects. Patient IV:4 had a normal cognitive profile, moderate depression, and anxiety. The psychological interview of patient III:3 also showed the presence of depression and anxiety.

Neurophysiological analysis was conducted in all affected females. There was no relief of peripheral neuropathy at nerve conduction studies (ENG) and electromyography (EMG). Motor evoked potentials (MEPs) of the lower limbs always showed bilateral, an asymmetric increase of central motor conduction time at lower limbs. Likewise, somatosensory evoked potentials (SEP) showed bilateral, an asymmetric increase of central conduction time at lower limbs. Brainstem auditory evoked potentials (BAEPs) were available for patients III:2 and IV:2. Patient III:2 showed a normal response. Patient IV:2 had bilateral and asymmetric dysfunctions of the auditory pathways. The visual evoked response was normal in all the analyzed cases (III:10; IV:2; IV:4).

Brain 1,5 tesla MRI in all affected female carriers showed some and small diffuse T2 hyperintensity localized in the context of the white matter of both cerebral hemispheres. Concomitant nuanced hyperintensity in the same sequences of the periventricular white matter were detected (Figure 2).

## 4. Discussion

As in many X-linked diseases, in adrenoleukodystrophy, it is expected that female carriers remain asymptomatic. Recent papers reported that most *ABCD1* female carriers indeed show at first instrumental abnormalities and, later on, frank neurological symptoms. Thus, what was once considered just a simple carrier, now develops a clinical phenotype reminiscent of a frank adrenomyeloneuropathy including cognitive impairment and motor disability. The natural disease progression of *ABCD1* heterozygous female carriers is strongly age-related since roughly 80% of the female carriers carry some symptom beyond the age of 60. Contrariwise to AMN where adult males show complete penetrance, female carriers show a reduced penetrance with a cut-off age at about 40 years critical for phenotypic conversion.

Data concerning female carriers have been produced by cross-sectional cohort studies or longitudinal data retrospectively reviewed. These data provided a detailed description of clinical and neurological phenotype in females but failed to focus on the genotype-phenotype correlation.

Conversely, in this paper, by profiting off a study that was conducted in a large family affected by X-ALD, such a correlation was addressed. In this family, all affected males died within the first decade. Female carriers are heterozygote for the mutation p.(Arg554His) in the *ABCD1* gene. This mutation is pathogenic considering the AMG classification and the disease-specific mutation database available at www.x-ald.nl, accessed on 10 October 2020.

This variant, mainly reported in cross-sectional analysis considering X-ALD male patients, is not described in female carriers, and genotype/phenotype has never been evaluated. Our systematic search of the literature conducted to identify carriers of p.(Arg554His) mutation shows that male patient carriers of this variant suffer from ALD, CCALD, and AMN (Table 2).

This is in line with other papers reporting the lack of genotype/phenotype in X-ALD correlation in males. The common frameshift mutation (p.Gln472fsX83) leading to a truncated ALDP identified in 81 patients, and the p.Pro484Arg identified in a family with six male patients, presented with a wide clinical variability of X-ALD [36,37].

Another paper by Margoni et al., 2017 [38] highlighted the wide range of phenotypic expressions of ALD, reporting a novel heterozygous mutation IVS4+2T>A (c.1393+2T>A) in a family with six members (two females) carrying different phenotypes of the entire clinical and radiological spectrum of X-ALD.

Females heterozygous for X-ALD can develop a wide range of neurologic abnormalities, most of them consisting of an AMN-like phenotype, and neurological impairment may finally affect most (if not all) of them and progress with age in severity, independently from the *ABCD1* variant [39].

Therefore, studies concerning large families of X-ALD considering genotype/phenotype in females are required.

In this report, we analyzed a large family affected by X-ALD, with five female carriers of p.(Arg554His) mutations and developing full penetrance within 42 years. All the women showed at first progressive spastic or ataxic-spastic gait, tetra-hyperreflexia, lower limbs hyposthenia, and sensory complaints. Evoked motor and somatosensory potentials were affected abnormally in all the female carriers, confirming the involvement of the CNS at the level of the spinal cord.

The neuropsychological assessment showed deficits in praxic-constructive and logical-deductive skills in one patient while a depression/anxiety spectrum was evident in all tested patients. Cognitive impairment and/or psychiatric symptoms in our female population suggest the presence of brain involvement in AMN-like phenotype.

While the neuroimaging findings of cerebral ALD and AMN have been well described in men, the same findings in women are less investigated [2]. We found a non-specific 1.5 T brain MRI pattern in all the symptomatic female carriers characterized by more or less diffuse T2 hyperintense lesions in white matter associated with nuanced periventricular hyperintensity. Although these represent non-specific and nuanced changes, the carriers did not have the common cardiovascular risk factors, so these lesions cannot be explained otherwise and must be taken into account.

This framework suggests giving more attention to routine brain MRI in suspected X-ALD carriers, also in AMN-like phenotype, to detect the slight and non-specific pattern and to better define neuroimaging findings in female carriers.

## 5. Conclusions

In this study, we describe a large family that allowed us to study the role of an *ABCD1* mutation in five female carriers that developed a full penetrance within 42 years. These allowed to strengthen the relevance of clinical symptoms in the female, which leads to a better understanding of the role of the genetic background.

Although a genotype-phenotype correlation in male patients has not been established, the data suggest the relevance of obtaining information concerning this association in female carriers.

Moreover, these data indicate the relevance to include *ABCD1* genes in genetic panels for gait disturbance in women.

## Figures and Tables

**Figure 1 genes-12-00775-f001:**
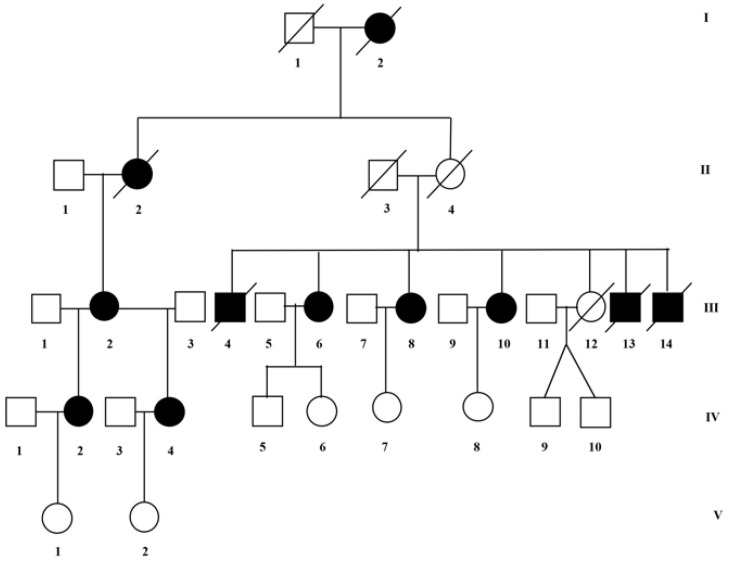
Pedigrees of the investigated family. The genetic evaluation included nine people, eight female and one male. Of the eight females, five are affected: III:2; III:6; III:10; IV:2; and IV:4, and three are healthy: IV:8; V:1; and V:2. The affected male died within the first decade (III:4; III:8; III:13; and III:14).

**Figure 2 genes-12-00775-f002:**
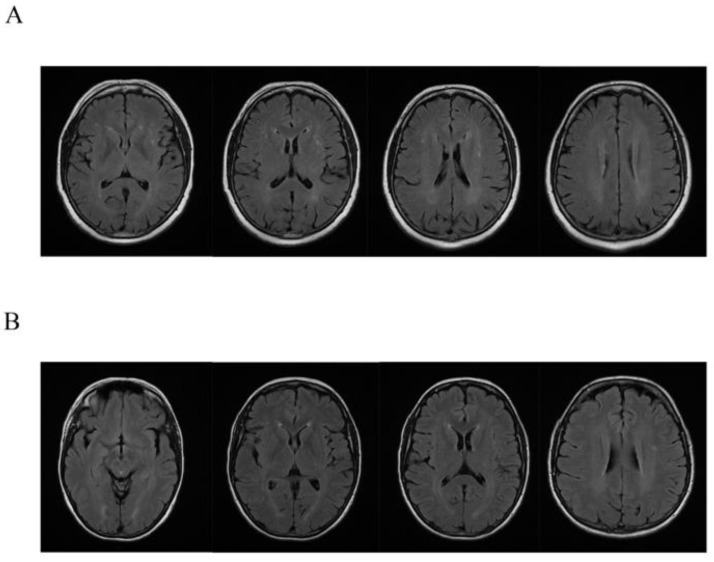
Axial FLAIR MRI sequences show small and diffuse hyperintensities in white matter and nuanced periventricular hyperintensities in patients III:6 (**A**) and IV:2 (**B**).

**Table 1 genes-12-00775-t001:** Clinical and biochemical features of the five symptomatic female carriers of the p.Arg554His (III:2; III:6; III:10; IV:2 and IV:4) and one healthy male. VLFA dosage was conducted in plasma. C26:0 levels, C24:0/C22:0 and C26:0/C22:0 ratios were increased in all the female samples (C26:0 levels * *p* = 0.0078; C26:0/C22:0 ** *p* = 0.0127).

	Symptoms of Onset	Age of Onset	Age at First Examination	Phenotype	Gait or Balance Difficulties	Sensory Complaints	Sphincter Disturbances	Psiciatric or Cognitive Impairment	Pyramidal Signs	Other Neurological Signs	Genotype	C26:0 (0.010–0.900)	C26:0/C22:0 (0.006–0.020)	C24:0/C22:0 (0.470–1.270)
III:2	Gait difficulties		68	AMN	Spastic	No	No	-	Yes	No	Heterozygote	1.29 *	0.025 **	1.1
III:6	Gait difficulties	45	54	AMN	Spastic	No	-	Anxiety and depression	Yes	Diplopia	Heterozygote	1.8 *	0.02 **	1.2
III:10	Gait difficulties	39	43	AMN	Spastic	Lower limb pain	-	-	Yes	No	Heterozygote	1.71 *	0.023 **	1.212
IV:2	Imbalance and falls	35	46	AMN	Ataxic-spastic	No	Yes	Anxiety and depression. A slight deficit in praxic-constructive and logical-deductive skills	Yes	Dysmetria	Heterozygote	1.27 *	0.027 **	1.4
IV:4	Paraesthesia in lower limbs	34	44	AMN	Ataxic-spastic	Lower limb paresthesia	Yes	Anxiety and Depression	No	Lower limb apallesthesia	Heterozygote	1.27 *	0.028 **	1.28
IV:8 (health)												0.59	0.01	0.835

**Table 2 genes-12-00775-t002:** This table shows 14 publications describing 23 carriers of p.(Arg554His) variant. The gender is reported for 13 male patients. The phenotype reported in association with this variant includes ALD (adrenoleukodystrophy), CCALD (childhood cerebral ALD), AMN (adrenomyeloneuropathy), and asymptomatic carrier. (NA = Not available).

Sample ID	Population	Gender	Age	Age Onset	Phenotype	References
61897C	USA	NA	NA	NA	ALD	[23]
71597A					Asymptomatic	
32597B					ALD	
1	USA	NA	NA	NA	NA	[20]
A161	Spain	NA	NA	NA	CCALD	[24]
A153					CCALD	
A344					CCALD	
A412					AMN	
AMN-C2	France	M	43	41	AMN	[25]
AMN-C4		M	NA	30	AMN	
AMN-C12		M	NA	31	AMN	
1	Italy	NA	NA	NA	NA	[26]
A12	China	M	NA	NA	CCALD	[27]
5	Japan	M	11	7	CCALD	[28]
1	Japan	NA	NA	NA	CCALD	[29]
9	South America	M	NA	NA	Asymptomatic	[30]
III:1	China	M	41	26	AMN	[31]
III:2		M	40	20	AMN	
P2	China	M	NA	10	CCALD	[32]
11	Korea	M	24	23	AMN	[33]
15	China	M	NA	NA	CCALD	[34]
1	China	M	NA	11	CCALD	[35]
2		M	NA	12	CCALD	

## Data Availability

https://sites.google.com/uniurb.it/repository/a-large-family-with-p-arg554his-mutation-in-abcd1-clinical-features-and-ge?authuser=0, accessed on 10 October 2020. The original image of NMR will be made available by the authors, without undue reservation.

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
