# Peer review of "A Large Family with p.Arg554His Mutation in ABCD1: Clinical Features and Genotype/Phenotype Correlation in Female Carriers"

_genes, 2021, doi:10.3390/genes12050775_

Round 1
Reviewer 1 Report
The work from Campopiano et at. is focused on the genetic study of a family carrying a pathogenic variant in the ABCD1, a gene associated with X-linked adrenoleukodystrophy. The mutation p.Arg554His showed a clear segregation with the pathological phenotype (spastic or ataxic-spastic gait, tetra-hyperreflexia, etc.).
The paper is well-written, and the conclusions are based on the results. The study helps to strengthens previous observation that women are not only carriers, but also could develop the disease with a late onset. I recommend the publication in the present form
Author Response
We would like to thank the reviewer
Reviewer 2 Report
The authors have submitted a manuscript of illustrating possible correlation between genotype and phenotype of the mutations of the adenosine triphosphate-binding cassette protein subfamily D (ABCD1) observed in a five-generation family in which segregates two main phenotypes of X-linked adrenoleukodystrophy (X-ALD). The authors described the relevance of clinical symptoms in female carriers of ABCD1 mutations which leads to better the importance of genetic information for the diagnosis. However, to better understand the usefulness of genetic analyses of ABCD1 mutations against X-ALD, readers need to easily know the take-home messages from this research article: as mentioned by the authors in the introduction, about 600 different mutation have been reported in the analysis of ABCD1 genotype and X-ALD phenotype. The authors should include an additional table which represents possible significant relationship between ABCD1 mutations and X-ALD phenotypes, because there is no definite description about this issue. The authors are strongly recommended to add the take-home message, especially the positive result-induced diagnostic prediction. Given published reports from other research group(s), the authors should describe how the ABCD1 product be affected by the specific mutation(s) which will lead to specific phenotype.
Author Response
We would like to thank the reviewer for the valuable suggestions.
We hope that our corrections fit with the reviewer requests.
thanks a lot

Reviewer 3 Report
1. InterVar website link is not working please provide correct link. 2. Table 1 doesn’t have any statistical tests being performed. 3. No mechanistic or genetic rigor tests been conducted in this study. 4. Methods used in this study is basic. 5. In Figure 2 it is hard to figure out small and diffuse hyperintensities in white matter and nuanced periventricular hyperintensities. The authors need to explain it clearly.Author Response
We would like to thank the reviewer for the valuable suggestions.
We hope that our corrections fit with the reviewer requests.
thanks a lot

Round 2
Reviewer 2 Report
The revised version of the manuscript has been improved adequately on the basis of the comments raised by reviewers, and thus it is now recommended to be accepted for publication in Genes.Reviewer 3 Report
The authors have added substantial changes and can be published in present form